# The Impacts of Standardized Flaxseed Meal (XanFlax) on the Physicochemical, Textural, and Sensory Properties of Muffins

**DOI:** 10.3390/foods12224085

**Published:** 2023-11-10

**Authors:** Ju Hui Lee, Youn Young Shim, Martin J. T. Reaney, Jin A Yoon

**Affiliations:** 1Department of Food and Nutrition, Gangseo University, Seoul 07661, Republic of Korea; 2Department of Food and Bioproduct Sciences, University of Saskatchewan, Saskatoon, SK S7N 5A8, Canada; younyoung.shim@usask.ca (Y.Y.S.); martin.reaney@usask.ca (M.J.T.R.); 3Prairie Tide Diversified Inc., Saskatoon, SK S7J 0R1, Canada; 4Department of Food and Biotechnology, Korea University, Sejong 30019, Republic of Korea

**Keywords:** flaxseed, *Linum usitatissimum* L., XanFlax, muffin, baking, sensory evaluation

## Abstract

Flaxseed is becoming increasingly popular as a superfood due to its many health benefits. While flaxseed is considered an oilseed, flaxseed meal (the by-product of flaxseed oil extraction) also contains many nutritional compounds not found in the oil. This study explored the use of a Canadian flaxseed (*Linum usitatissimum* L.) meal product to fortify bakery foods and improve their nutritional properties. Muffins were made using a control recipe as well as four different formulations that included varying amounts of a standardized flaxseed meal supplement called XanFlax (5, 10, 20, and 40%). The physicochemical properties of the muffins, including their texture, color, sugar content, pH, specific gravity, loss rate, and moisture, were evaluated. Additionally, the sensory attributes contributing to muffin quality were thoroughly examined. The lightness (*L**) and yellowness (*b**) of the muffins, which were highest in the control group at 82.22 and 34.69, respectively, decreased as the amount of XanFlax increased (*p* < 0.05). Additionally, the redness (*a**) of the muffins increased as the amount of XanFlax increased (*p* < 0.05). The muffins’ sugar content (2.00 brix%) remained consistent across all treatments and controls except for those prepared with 20% XanFlax (2.17 brix%). As the amount of XanFlax powder increased, the pH of the muffins increased significantly. The moisture content in the muffins was highest at 23.71 ± 0.79% in the 10% XanFlax treatment and lowest at 22.06 ± 0.30% in the 40% XanFlax treatment. The muffins enriched with 5% XanFlax had an average height of 5.35 cm and volume of 131.33 mL, surpassing the results for the muffins made with other formulas (*p* < 0.05). Additionally, the cohesiveness and gumminess of the muffins tended to increase with the addition of XanFlax. The most favorable attributes, namely the appearance, flavor, taste, texture, and overall acceptance, were consistently associated with the 5% and 10% XanFlax treatments (*p* < 0.05). This study marks the first time a standardized flaxseed gum product, XanFlax, has been described in a functional baking application.

## 1. Introduction

Due to the increasing interest in healthy functional foods and the demand for simple and nutritious meals, the use of superfood ingredients has become more diverse. There is a growing need for the development of healthy functional foods using natural ingredients. This trend has led to an increase in demand for healthier confectionery and baked goods [1]. The baking industry has evolved due to various factors such as economic changes and consumer needs, and products with enhanced health functionality are driving the market, reflecting the needs of health-conscious consumers [2,3]. Muffins are popular due to their soft texture and rich taste [4], and they are widely consumed as a snack or meal because of their simple and convenient preparation method. As people become more health-conscious, the demand for healthy functional foods that have physiological effects is also increasing. Studies have been conducted on muffins with enhanced health functionalities, such as those containing Aronia powder [5] and tangerine peel powder [6], among others. Research on the quality characteristics of muffins with various ingredients added has been ongoing, including muffins with mulberry concentrate [7], coffee ground extract and powder [8], various types of rice flour replacing wheat flour [9], ‘Fuji’ apple peel powder [10], and barley water [11].

Flaxseed (*Linum usitatissimum* L.) is a rich source of nutrients that are important for the modern diet, including protein, minerals, omega-3 fatty acids, lignans, dietary fiber, vitamins, and tocopherols [12]. It is also considered a high-protein food due to its high protein content (20%) and energy density level (4.5 kcal/g). However, flaxseed contains cyanogenic compounds that release cyanide, which is toxic. It is desirable to reduce or remove these compounds before using flaxseed meal products in food [13]. Flaxseed lignans inhibit cancer development and proliferation and have antioxidant effects [14]. In fact, flaxseed contains 75–800 times more lignans than cereals, legumes, fruits, and vegetables [15]. While secoisolariciresinol is the main flaxseed lignan, other lignans, including matairesinol, pinoresinol, lariciresinol, and isolariciresinol, are also present in small amounts [16]. Orbitides known as linusorbs are present in flaxseed, and 45 different linusorbs have been described [12]. Among them, [1–9-NαC]-linusorb B3 has antioxidant and anti-inflammatory effects. It also can inhibit the formation of the actin cytoskeleton, which is important for cancer progression, invasion, and metastasis and has in vitro anti-cancer effects, where it delays the migration of C6 cells [17]. Recently, Lee et al. [18] found that flaxseed-derived XanFlax is an excellent substitute for commercial thickeners such as Xanthan gum, guar gum, corn starch, and gluten-free flour, and can also be used as an egg substitute in various baked goods [18,19]. In addition to its physical thickening properties, this flaxseed-derived high-fiber product lowers the cholesterol and blood triglyceride levels in diabetic and control subjects [20,21]. XanFlax is a flaxseed meal derived from the cold pressing of flaxseed that is standardized so that it is an effective high-viscosity thickener, gelling agent, texture modifier, suspending agent, and stabilizer. As a flaxseed meal, it is an excellent source of omega-3, lignan, orbitides, tocopherol, protein, and dietary fiber, among others. Thus, it can add to the nutritional benefits or physicochemical “thickener” properties of flaxseed [18].

Flaxseed, traditionally used for medicinal purposes, has gained popularity in recent years as a functional food ingredient due to its nutritional and pharmacological benefits. However, the research on its utilization in confectionery and baking remains limited. The goal of this study was to increase the utility of this standardized flaxseed gum product, XanFlax (high-fiber flaxseed powder), in functional baking applications by making the first attempt to use it as an ingredient in muffin baking. The physicochemical quality characteristics and acceptability of the formulated muffins were investigated, and the results of this study could provide experimental data necessary for the development of new products with this ingredient.

## 2. Materials and Methods

### 2.1. Materials

The XanFlax powder, an additive or ingredient used for formulating muffins in this study, was provided by Prairie Tide Diversified Inc. (Saskatoon, SK, Canada). Table 1 presents the information presented on the nutritional facts label detailing the constituents of the product “XanFlax”. Other muffin ingredients were purchased from commercial marts and used in the preparation of muffins. The brands used were Sajodongaone Co., Ltd. (Seoul, Republic of Korea), for refined soft wheat flour (white); Farming Association Haemil Co., Ltd. (Yeoju, Republic of Korea), for fresh whole eggs; Beksul (Seoul, Republic of Korea) for white sugar; Hanju Salt Co., Ltd. (Ulsan, Republic of Korea), for refined salt; Galimfood Co., Ltd. (Incheon, Republic of Korea), for unsalted butter and baking powder; and Seoul Milk (Seoul, Republic of Korea) for non-fat dry milk.

### 2.2. Preparation of Muffins

An optimal muffin formulation was previously described by Song et al. [9], in which the mixing of ingredients was achieved by modifying the creaming method. XanFlax was added to the muffins in various concentrations of 0, 5, 10, 20, and 40% by partially substituting the total amount of 200 g of refined soft wheat flour (Table 2). Using a kneader (Horno Panadero, BS-201, Busung, Bucheon, Republic of Korea), butter and sugar were mixed for 1 min in the first stage, then increased to the second stage and mixed for 1 min more. Then, the melted butter and sugar were added in three portions to prevent the egg from separating from the rest of the mix. The cream was added in three portions and mixed for a total of 3 min to maintain the shape of the cream. Refined soft wheat flour, XanFlax powder, refined salt, and baking powder were then sieved through a 35-mesh sieve and added to the mixed cream and mixed for 1 min in the first stage of the kneading. Finally, milk was added and remixed in the first stage for 1 min to complete the dough. The finished dough was poured into tinfoil cups and lined with parchment paper (Φ60 mm × 45 mm), divided into 5 portions of 70 g each, and baked in an electric oven (Horno Panadero, BS-023, Busung, Bucheon, Republic of Korea) preheated to 175 °C for 25 min. The baked muffins were cooled at room temperature (22 °C) for 1 h and then used as samples.

### 2.3. Weight, Height, and Volume Measurements

The muffins with added XanFlax powder at concentrations of 0, 5, 10, 20, and 40% were evaluated. The height of the muffins was measured three times with a vernier caliper (150 × 0.05 mm, Eagle Vernier Caliper, Beijing, China) by cutting the center of the muffin and measuring the top height from the bottom of the cross-section. The volume of the muffins was measured three times using the seed displacement method with white rice [22] and the results are expressed as the average value.

### 2.4. Measurements of Specific Gravity and Baking Loss of Dough

The specific gravity of the dough was determined according to the AACC method 26–10 [22] by comparing the weight of the muffin dough to the weight of water. The baking loss of the muffins was calculated by measuring the weight of the dough before baking and the weight of the muffin after baking using an electronic balance (IB-410, Ohaus, Seoul, Republic of Korea). The difference between the two values was then calculated using Equations (1) and (2):(1)$$Specific\;gravity=\frac{Density\;of\;muffin\;dough\;(\frac{g}{mL})}{Density\;of\;water\;(\frac{g}{mL})}$$
(2)$$Baking\;loss\;rate\;(\%)=\frac{Weight\;of\;muffin\;dough\;(g)-Weight\;of\;muffin(g)}{Weight\;of\;muffin\;dough\;(g)}\times 100$$

### 2.5. Moisture Content, Sugar Content, and pH Measurement

The moisture content of the muffins was determined in triplicate using the AOAC method [23] by heating and drying 5 g of a powdered sample at 105 °C in a drying oven (KC0-150, Kuk Je Eng Co., Goyang, Republic of Korea) to a constant weight. After cooling in a vacuum desiccator (BA.42025-0000, Bel-Art Scienceware, NJ, USA) for 30 min, the weight was measured with an electronic balance (IB-410, Ohaus, Seoul, Republic of Korea). To measure the sugar content (%), 5 g samples at concentrations of 0, 5, 10, 20, and 40% were suspended in 50 mL of distilled water, filtered through an 8 μm paper filter (1002-150, Whatman, Incheon, Republic of Korea), and measured three times with a saccharimeter (Pocket Refractometer, Atago Co., Ltd., Tokyo, Japan). The pH values of the dough and muffins were also measured three times using a pH meter (pH7110, Inolab Especialistas, Mexico City, Mexico) for a solution obtained by adding 5 g of each sample to 50 mL of distilled water and filtering the suspension through an 8 μm paper filter.

### 2.6. Chromaticity and Section Measurements

The color of the muffins was analyzed using a colorimeter (CR-400, Konica Minolta Sensing Inc., Osaka, Japan) to measure the *L** value (lightness), *a** value (redness), and *b** value (yellowness) according to the Hunter scale. Three measurements were taken for each sample, with a white calibration plate used as the standard for comparison. The exterior cross-sections were then photographed and examined using a camera (LG wing, LG, Seoul, Republic of Korea) for a further analysis.

### 2.7. Texture Measurement

Following the method of Yoon [24] with some modifications, the texture of the muffins was evaluated using a texture analyzer (TA.XT *plus*/50 Stable Micro Systems, Godalming, UK) with a cylindrical probe of 75 mm diameter and a wide bottom plate. The sample (2 × 2 × 2 cm) was collected from the center of the muffin. The trigger force and test distance were 5 g and 10 mm, respectively. The pre-test and post-test speeds were both 2.0 mm/s and the test speed was 1.0 mm/s. The texture was measured using seven parameters: hardness, adhesiveness, springiness, cohesiveness, gumminess, chewiness, and resilience. The measurements were taken five times for each concentration of 0, 5, 10, 20, and 40% of XanFlax powder, and the average and standard deviation were recorded.

### 2.8. Sensory Evaluation

A total of 35 assessors (25 women and 10 men) aged between 21 and 24 years (mean age: 22 ± 2 years) were selected from the students of the Department of Food and Nutrition of Gangseo University. Only subjects who were regular muffin consumers (muffin consumption corresponding to 1–2 muffins per week) were involved in the study. None of the participants had previous or present taste or smell disorders. Informed consent was obtained from all subjects. The selected assessors majoring in food and nutrition had the ability to perform sensory tasks and had interacted in discussions of sensory attributes during six 1-h training sessions conducted to train the assessors to be familiar with the muffin products and the prescribed sensory evaluation procedures related to taste and tactile sensations. The training sessions were performed in a collective room and sensory booths of the Department of Food and Nutrition (Gangseo University, Seoul, Republic of Korea), designed under ISO guidelines [25]. The assessors were asked to rate their acceptability using a 7-point scale, with 1 indicating “dislike very much”, 4 indicating “average”, and 7 indicating “like very much”. The muffins were cut into 2 × 2 × 2 cm pieces and served on a white plate, and the participants were instructed to rinse their mouths with water after trying each sample. The acceptance test evaluated five attributes: appearance, flavor, taste, texture, and overall acceptability [24].

### 2.9. Statistical Anaysis

All statistical analyses were performed using the Statistical Package for the Social Sciences (SPSS) version 18.0 (IBM SPSS Inc., Chicago, IL, USA). Mean comparisons were made using a one-way analysis of variance (ANOVA) followed by Duncan’s multiple range test. The data are presented as means ± standard deviations (SDs) (*n* = 3), and *p* < 0.05 was considered statistically significant.

## 3. Results and Discussion

### 3.1. Measuring the Baking Properties of Muffins Prepared with XanFlax Powder and Observing Their Appearance

Table 3 illustrates the weight, height, and volume of muffins made with varying levels of XanFlax powder. The weights of the muffins were similar across all groups, with mean weights of 64.72 ± 0.50 g for the control group and 65.21 ± 0.58 g, 65.24 ± 0.40 g, 65.22 ± 0.24 g, and 65.09 ± 0.38 g for the muffins with 5, 10, 20, and 40% XanFlax powder added, respectively. This was likely because the addition of XanFlax powder did not have a notable impact on the weights of the muffins [6]. The heights of the muffins also varied, with the highest height of 5.35 ± 0.05 cm in the 5% added group and the lowest of 4.68 ± 0.08 cm in the 40% XanFlax powder group, while the other groups showed insignificant differences. As the concentrations of XanFlax powder increased to 5% and 10%, the muffin volumes increased to 131.33 ± 4.20 mL and 123.33 ± 5.20 mL compared to the control group (120.77 ± 2.66 mL). However, the volumes gradually decreased to 116.33 mL and 100.20 ± 1.92 mL in the 20% and 40% groups, respectively. This trend was similar to previous studies on muffins made with added *Suaeda japonica* Makino powder [24] and flaxseed [26], which showed that the volume decreased as the added material increased due to the gluten-free properties of the flaxseed. The reduced volume can be attributed to the gluten-free properties of flaxseed, which may affect the structural integrity and gas-trapping ability during baking, resulting in more compact muffin structures.

Table 3 also shows the results of measuring the specific gravity and baking loss rate values of muffins with varying levels of XanFlax powder. The specific gravity values of muffins with 0, 5, 10, 20, and 40% of XanFlax powder added were 0.41, 0.41, 0.41, 0.41, and 0.43, respectively. The values significantly increased as the amount of XanFlax powder added increased (*p* < 0.05). This observation suggests that the incorporation of XanFlax powder influenced the overall density and structure of the muffins. Notably, a higher specific gravity indicates a decrease in air content within the muffin, leading to the formation of smaller pores and potentially resulting in muffins with poorer chewiness. On the other hand, a lower specific gravity would imply an increased air content, which may contribute to improved chewiness of the muffin [27,28]. The baking loss rates were 7.55 ± 0. 72%, 6.84 ± 0. 83%, 6.81 ± 0.57%, 6.83 ± 0.35, and 7.01 ± 0.54, respectively, showing no significant difference. The baking loss occurs when heat is applied to the dough and the vapor pressure increases, causing liquid with a low boiling point to expand and escape as gas [29]. The similar baking loss rates suggest that the addition of XanFlax powder did not substantially affect the vaporization and loss of volatile components during the baking process, which is an important factor to consider in the overall texture and quality of the final product. This observation suggests that XanFlax powder may not substantially impact the dough’s consistency, texture, or moisture-retaining properties.

### 3.2. Measurements of Moisture Content, Sugar Content, and pH of Muffins

Table 4 shows the results in terms of the measured moisture content, sugar content, and pH values of muffins with varying levels of XanFlax powder. The moisture contents of the muffins were 24.17 ± 0.39% in the control group, 23.31 ± 0.60% in the 5% XanFlax group, 23.71 ± 0.79% in the 10% group, 22.35 ± 0.60% in the 20% group, and 22.06 ± 0.30% in the 40% group. As the concentration of XanFlax powder increased, the moisture content of the muffins tended to decrease (*p* < 0.05). Previous studies have shown a direct relationship between the dietary fiber content and moisture content, with a reported moisture content of 14.02% [30]. Therefore, it is anticipated that the binding force between the moisture and XanFlax powder increased due to the high dietary fiber content in XanFlax, leading to a decrease in moisture content as the ratio of bound water increased relative to the control [31]. This variation in the water retention abilities of the samples, depending on the dietary fiber and its water-binding capacity, is thought to have affected the water content of the muffins [32]. The sugar contents of the doughs exhibited no significant differences among the different groups, although interestingly the sugar content of the muffins was the highest with 40% XanFlax powder (2.17 ± 0.14, Brix%) (*p* < 0.05). This finding suggests that the sugar content of the XanFlax powder may have influenced the overall sugar content of the muffins. Although the exact mechanism behind this observation requires further investigation, it is reasonable to assume that the inherent sugar content of the XanFlax powder contributed to the elevated sugar levels in the muffins. There were no significant differences in the pH levels of the doughs, although the pH of the muffins increased as XanFlax powder was added. The pH of the muffins was the lowest in the control group at 6.31 ± 0.06, and as the amount of XanFlax powder increased, the pH of the muffins increased significantly (*p* < 0.05). The pH of the muffins was influenced by the pH and moisture content of the added ingredients [24], and in this case, the pH increased because of the pH and moisture content of the XanFlax powder.

### 3.3. Chromaticity and Cross-Section Appearance

The lightness (*L**) and yellowness (*b**) values of the muffins were the highest (82.22 ± 0.85 and 34.69 ± 0.86, respectively) in the control group (Table 5). These values tended to decrease as the amount of XanFlax powder increased (*p* < 0.05). Relatively low redness (*a**) values (–2.29 ± 0.21 and 1.54 ± 0.17) were observed in the control group and the 5% added group, which increased significantly as the amount of XanFlax powder increased (*p* < 0.05). The color change was attributed to the dark brown color of the XanFlax. Various factors influence the color of muffins, including the type of flour, the quality and quantity of ingredients used, and the temperature and time of baking [33]. The decrease in *L** value and increase in *a** value were due to the inherent dark brown color of the flaxseed, the main component of XanFlax. This finding was consistent with previous studies, wherein the amount of added material increased and the color of the material became darker, affecting the *L**, *a**, and *b** values [25,32,34]. This effect can be attributed to the higher concentration of XanFlax powder, which introduces more dark brown pigments into the muffin batter, resulting in the observed color shifts. The color changes in muffins due to the incorporation of XanFlax powder may affect consumer perceptions and their acceptability. While some consumers may appreciate darker, richer colors as a sign of the addition of natural ingredients, others may prefer lighter-colored muffins. Therefore, further research could explore the sensory aspects of these supplemented muffins to assess the impact of supplementation on the overall product’s appeal and acceptability.

As the concentration of XanFlax powder increased, the muffins appeared to rise less (Figure 1). A cross-sectional examination of the muffins revealed that the internal structure of the muffins in the additive group was denser with irregularly distributed pores as compared to the control group. A similar result was also reported in a study on muffins made with added kaniwa powder [32]. The substitution of gluten in the gluten network with dietary fiber in XanFlax powder seems to weaken the three-dimensional network structure in the dough [32]. This results in the volume of the muffin being reduced and the gluten dilution effect interfering with the formation of an optimal gluten network, leading to small, irregular pores. Thus, the addition of high-fiber XanFlax powder may improve the overall quality of the muffins by directly impacting external factors such as the volume and height, as well as the interior structure.

### 3.4. Texture Properties

The texture profile analysis (TPA) of the muffins made with XanFlax powder was conducted by evaluating seven parameters: hardness, adhesiveness, springiness, cohesiveness, gumminess, chewiness, and resilience (Table 6). The TPA provided valuable insights into how the XanFlax powder influenced the muffins’ texture attributes.

The hardness of the muffins increased as the concentration of XanFlax powder increased, with the 5% group having a value of 486.31 ± 53.66 g, the 10% group having a value of 502.29 ± 73.22 g, the 20% group having a value of 728.95 ± 57.07 g. and the 40% group having a value of 831.23 ± 31.82 g (*p* < 0.05). A similar pattern was observed in a study using flaxseed as part of a muffin recipe [35]. In a study of muffins that incorporated peach dietary fiber, it was reported that the muffin hardness increased with the substitution of flour with greater amounts of peach fiber [36]. Based on these reports, it is possible that the increase in muffin hardness with the addition of increasing amounts of XanFlax is due to the dilution of gluten [37]. It was observed that as the concentration of XanFlax powder increases, the volume of the muffin becomes smaller and the internal structure becomes denser, resulting in a harder texture.

The adhesiveness was highest in the control group (−0.59 ± 0.08 gs) and gradually decreased as the amount of XanFlax powder increased, being the lowest in the 40% group (−2.33 ± 0.09 gs) (*p* < 0.05). Regarding the cohesiveness, the highest score was found in the 10% XanFlax powder group (1.01 ± 0.07%) (*p* < 0.05), and a similar pattern was also found in a study by Jang et al. [38]. Although there was no significant difference in cohesiveness between the 20% and 40% XanFlax powder added groups, it significantly decreased (*p* < 0.05). A decrease in adhesiveness of the XanFlax muffins with respect to the controls could arise from several factors. First, the muffins would have an elevated fat content, and the fat could lubricate their surfaces. Second, the flaxseed muffins would contain less gluten and the this could produce a less adhesive product. Third, flaxseed polymers form coacervates with proteins, and the soluble fiber of the flaxseed might coat the protein particles in the muffin. Similarly, the decreased cohesiveness with increasing XanFlax contents can be attributed to the high soluble dietary fiber content of XanFlax and the formation of coated protein in the muffin recipe. The soluble fiber also affects the muffins’ water binding properties. Similarly, the higher gumminess and chewiness levels in muffins with higher XanFlax concentrations were consistent with the influence of dietary fiber on these texture attributes. The ability of XanFlax to impact these textural characteristics is similar to what has been observed in studies pertaining to tangerine peel powder [6] and rice-bran-enriched muffins [38].

The springiness levels were similar between the control group and the XanFlax powder-added groups. A similar trend was found in muffins made with tangerine peel powder and domestic blueberries [6,39]. In another study using black rice powder [40], the cohesiveness and springiness of the muffins gradually decreased as the amount of added black rice powder increased. Similarly, the elasticity of the muffins was significantly reduced upon the addition of corn to the mix [41]. On the other hand, the springiness levels of the muffins showed no significant difference between the control group and the XanFlax powder-added groups, indicating that the elasticity of the muffins’ structure was not substantially affected by the XanFlax incorporation. This contrasts with findings from studies involving black rice powder [40] and corn [41], which reported a gradual decrease in springiness with increasing amounts of added materials.

The gumminess and chewiness were the highest in the 40% XanFlax powder group (981.52 ± 86.09 g/s and 983.14 ± 62.05 g, respectively) (*p* < 0.05). It was found that the high content of dietary fiber in the XanFlax affected the adhesiveness and adhesion as the concentration increased. This pattern is similar to previous reports of muffins produced with added tangerine pericarp [6] or added rice bran powder [38].

The resilience, which indicates the property of returning to the original shape and form after pressing the muffin, was twice as high in the control group (0.24 ± 0.04 g) as compared to the XanFlax powder groups (*p* < 0.05), and there were no significant differences between the groups with added XanFlax.

These contradictory results can be attributed to the properties of the added ingredients. As the amount of XanFlax powder increases, the cooked appearance in the muffins increases. The structure becomes denser, leading to increased hardness, gumminess, and chewiness [34]. However, when XanFlax powder was added at the 5% level, this resulted in the production of muffins with the most desirable texture properties, which were soft and moist.

### 3.5. Sensory Evaluation

The sensory characteristics of muffins made with XanFlax powder were analyzed in terms of their appearance, flavor, taste, texture, and overall acceptance, as shown in Table 7. The appearance of the control group muffins (6.08 ± 1.08) was considered the best, with no significant differences among the muffins containing XanFlax powder.

The flavor score was highest (5.22 ± 1.27) in the group with 5% XanFlax powder and lowest (4.35 ± 1.76) in the group with 40% XanFlax powder. The flavor score decreased as the XanFlax content increased (*p* < 0.05). A similar trend was also observed in terms of the taste scores, where the control and the 5% and 10% XanFlax powder groups scored higher (5.25 ± 1.44, 4.89 ± 1.30, and 5.19 ± 1.24, respectively) but the groups with 20% and 40% XanFlax powder scored lower (4.00 ± 1.52 and 3.97 ± 1.73, respectively) (*p* < 0.05).

The scores for texture were highest in the groups with 5% and 10% XanFlax powder (5.08 ± 1.05 and 5.17 ± 1.16, respectively) and lowest in the group with 40% XanFlax powder (4.24 ± 1.81) (*p* < 0.05). In terms of the overall acceptability, the group with 5% XanFlax powder had the highest score (5.00 ± 1.35) and the group with 40% XanFlax powder had the lowest score (4.06 ± 1.86).

Apart from their appearance, the 5% and 10% XanFlax powder muffins were the most acceptable, while the 40% muffins had the lowest acceptability. Similar changes in rheological properties were reported in previous studies involving the addition of whole flaxseed powder to sponge cake [42,43]. Therefore, it is concluded that using XanFlax powder in the confectionery industry at concentrations of 5% to 10% is likely to produce muffins of the highest overall quality.

## 4. Conclusions

In this study, we evaluated the impacts of varying levels of XanFlax powder on the quality attributes of muffins. Our analysis provided valuable insights into the physical, chemical, and sensory characteristics of muffins across a range of XanFlax concentrations, from 0% to 40%. Notably, the muffin weights remained similar, indicating that the incorporation of XanFlax powder did not significantly affect the overall mass. However, our investigation revealed significant variations in other critical parameters. Muffins with 5% XanFlax powder exhibited increased height and volume, showing promise in terms of product development and consumer appeal. Additionally, the specific gravity was notably higher in the muffins containing 40% XanFlax powder, suggesting potential advantages in specific applications. While no significant differences were observed in terms of the baking loss rate, moisture content, dough sugar content, or pH, there were substantial changes in the muffin characteristics with increasing XanFlax powder levels. Notably, the color darkened as the XanFlax powder content increased, affecting the lightness (*L** value), yellowness (*b** value), and redness (*a** value). These findings underscore the importance of visual appeal when utilizing XanFlax powder. The texture analysis revealed that the hardness, cohesiveness, gumminess, and chewiness increased, while the restoration and adhesion decreased with higher XanFlax powder levels. The elasticity remained relatively consistent, offering insights into the desired mouthfeel and sensory experiences of the muffins. In taste tests, the appearance ratings were highest in the control group, with no significant differences in taste among the XanFlax groups. This suggests that XanFlax powder can be added to muffin recipes without compromising the taste. In conclusion, XanFlax powder holds significant potential to impact the quality of muffins. The sensory test data demonstrate that its incorporation leads to distinct changes in the sensory characteristics and overall acceptability of muffins. As the XanFlax content increased, we observed a gradual decline in the appearance, flavor, taste, and texture ratings, ultimately affecting the muffins’ overall acceptability. These findings underline the need for a nuanced approach when utilizing XanFlax powder in bakery product development. The unique properties of XanFlax, including its water-soluble gums, play a significant role in influencing these sensory attributes. To further optimize its application in bakery product development, future studies should explore its interactions with various ingredients in muffin formulations. Investigating its synergistic effects will help identify ideal use cases and applications for this promising ingredient. It is important to acknowledge that this study did not explore the full spectrum of XanFlax powder concentrations and possible adjustments to the recipe. Therefore, future studies could seek to balance the health benefits of flaxseed and the sensory preferences of consumers. Additionally, further research studies could investigate potential ingredient modifications or complementary flavors to mitigate the sensory changes associated with higher XanFlax powder contents. Considering consumer acceptance and nutritional analyses in future research will provide a more holistic perspective of XanFlax powder’s incorporation in bakery products. These analyses could help strike a balance between the health benefits associated with flaxseed and the sensory expectations of consumers, ultimately making bakery products enriched with XanFlax more appealing to a broader audience. This research marks the first time that XanFlax, a standardized flaxseed flour containing water-soluble gums, has been reported in functional baking applications. It is expected to contribute to the ongoing innovation and enhancement of baked goods while meeting the evolving demands of both consumers and the food industry. In summary, while our current study sheds light on the initial effects of XanFlax powder on the quality of muffins, it opens the door for continued research in this area. These findings provide valuable insights for the food industry in developing nutritious and palatable baked goods. The limitations highlighted and future perspectives outlined in this discussion are essential for fine-tuning the use of XanFlax powder in muffin production and addressing the multifaceted challenges and opportunities it presents.

## Figures and Tables

**Figure 1 foods-12-04085-f001:**
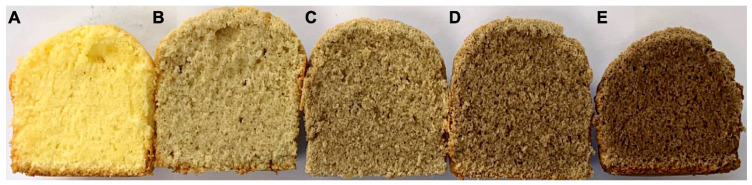
Muffins with various levels of supplemental XanFlax powder: control (flour without XanFlax powder) (**A**); flour with 5% XanFlax powder (**B**); flour with 10% XanFlax powder (**C**); flour with 20% XanFlax powder (**D**); flour with 40% (**E**) XanFlax powder.

**Table 1 foods-12-04085-t001:** XanFlax nutritional facts label.

Nutrition Facts		Serving Size 4 Tbsp (30 g)
*Calories*		141 ^1^
*Total Fat*		5.4 g
	Polyunsaturated fat	3.8 g
	Monounsaturated fat	1.1 g
	Saturated fat	0.7 g
	Trans fat	0.0 g
	Cholesterol	0.0 mg
*Total Carbohydrate*		10.1 g
	Dietary fiber	7.4 g
	Crude fiber	1.4 g
*Protein*		10 g

^1^ Amount per serving.

**Table 2 foods-12-04085-t002:** Formulations for the muffins made with XanFlax powder.

Ingredients (g)	XanFlax Content in Mix with Refined Wheat Flour (%)				
	0 ^1^	5	10	20	40
Refined soft wheat flour	200	190	180	160	120
XanFlax powder	0	10	20	40	80
White sugar	130	130	130	130	130
Unsalted butter	100	100	100	100	100
Whole egg	100	100	100	100	100
Non-fat dry milk	100	100	100	100	100
Baking powder	4	4	4	4	4
Refined salt	1	1	1	1	1

^1^ 0%: Flour without XanFlax powder (control).

**Table 3 foods-12-04085-t003:** Baking properties of muffins prepared with XanFlax powder.

Property	XanFlax Content in Mix with Refined Wheat Flour (%)				
	0	5	10	20	40
Weight (g)	64.72 ± 0.50 ^a^	65.21 ± 0.58 ^a^	65.24 ± 0.40 ^a^	65.22 ± 0.24 ^a^	65.09 ± 0.38 ^a^
Height (cm)	5.23 ± 0.08 ^b^	5.35 ± 0.05 ^a^	5.22 ± 0.10 ^b^	5.20 ± 0.11 ^b^	4.68 ± 0.08 ^c^
Volume (mL)	120.77 ± 2.66 ^bc^	131.33 ± 4.20 ^a^	123.33 ± 5.20 ^b^	116.33 ± 3.88 ^c^	100.20 ± 1.92 ^d^
Specific gravity (unitless)	0.41 ± 0.00 ^b^	0.41 ± 0.00 ^b^	0.41 ± 0.00 ^b^	0.41 ± 0.00 ^b^	0.43 ± 0.00 ^a^
Loss rate (%)	7.55 ± 0.72 ^a^	6.84 ± 0.83 ^a^	6.81 ± 0.57 ^a^	6.83 ± 0.35 ^a^	7.01 ± 0.54 ^a^

Values (means ± SDs) within rows followed by different lower-case superscript letters are significantly different at *p* < 0.05 as shown by Duncan’s multiple range test.

**Table 4 foods-12-04085-t004:** Characteristics of muffins prepared with XanFlax powder.

Characteristic	XanFlax Content in Mix with Refined Wheat Flour (%)				
	0	5	10	20	40
Moisture (Baking, %)	24.17 ± 0.3 ^a^	23.31 ± 0.6 ^b^	23.71 ± 0.79 ^ab^	22.35 ± 0.60 ^c^	22.06 ± 0.30 ^c^
Sugar (Dough, Brix%)	1.67 ± 0.29 ^a^	1.78 ± 0.29 ^a^	1.83 ± 0.14 ^a^	1.90 ± 0.25 ^a^	2.00 ± 0.25 ^a^
Sugar (Baking, Brix%)	2.00 ± 0.00 ^b^	2.00 ± 0.00 ^b^	2.00 ± 0.00 ^b^	2.00 ± 0.00 ^b^	2.17 ± 0.14 ^a^
pH (Dough)	5.73 ± 0.05 ^a^	6.05 ± 0.47 ^a^	6.06 ± 0.47 ^a^	5.75 ± 0.56 ^a^	6.11 ± 0.64 ^a^
pH (Baking)	6.31 ± 0.06 ^d^	6.88 ± 0.18 ^b^	6.71 ± 0.03 ^c^	6.98 ± 0.13 ^ab^	7.13 ± 0.00 ^a^

Values (means ± SDs, *n* = 3) in each row followed by different lower-case superscript letters are significantly different at *p* < 0.05 as shown by Duncan’s multiple range test.

**Table 5 foods-12-04085-t005:** Variation of color characteristics in muffins with varying levels of XanFlax powder addition.

Color	XanFlax Content in Mix with Refined Wheat Flour (%)				
	0	5	10	20	40
*L**	82.22 ± 0.85 ^a^	67.93 ± 0.74 ^b^	62.11 ± 1.48 ^c^	54.89 ± 1.10 ^d^	47.85 ± 1.92 ^e^
*a**	–2.29 ± 0.21 ^e^	1.54 ± 0.17 ^d^	3.61 ± 0.26 ^c^	5.41 ± 0.20 ^b^	6.87 ± 0.09 ^a^
*b**	34.69 ± 0.86 ^a^	21.09 ± 0.45 ^b^	18.20 ± 0.06 ^c^	16.67 ± 0.15 ^d^	15.02 ± 0.42 ^e^

Values (means ± SDs) within rows followed by different lower-case superscript letters are significantly different at *p* < 0.05 by Duncan’s multiple range test. *L**, brightness/darkness; *a**, (+) redness/(−) greenness; *b**, (+) yellowness/(−) blueness.

**Table 6 foods-12-04085-t006:** Texture properties of muffins made with XanFlax powder.

Property	XanFlax Content in Mix with Refined Wheat Flour (%)				
	0	5	10	20	40
Hardness (g)	852.25 ± 93.18 ^a^	486.31 ± 53.66 ^b^	502.29 ± 73.22 ^b^	728.95 ± 57.07 ^ab^	831.23 ± 31.82 ^a^
Adhesiveness (gs)	−0.59 ± 0.08 ^a^	−0.45 ± 0.32 ^a^	−1.14 ± 0.25 ^ab^	−1.38 ± 0.19 ^ab^	−2.33 ± 0.09 ^b^
Springiness (%)	1.00 ± 0.01 ^a^	1.00 ± 0.00 ^a^	1.00 ± 0.00 ^a^	1.00 ± 0.00 ^a^	1.00 ± 0.00 ^a^
Cohesiveness (%)	0.77 ± 0.02 ^c^	0.91 ± 0.03 ^b^	1.01 ± 0.07 ^a^	0.88 ± 0.09 ^b^	0.98 ± 0.04 ^ab^
Gumminess (g/s)	833.32 ± 98.70 ^b^	512.63 ± 79.47 ^c^	583.59 ± 75.87 ^c^	862.00 ± 85.62 ^ab^	981.52 ± 86.09 ^a^
Chewiness (g)	837.49 ± 93.06 ^b^	512.63 ± 78.92 ^c^	583.59 ± 76.90 ^c^	862.00 ± 84.95 ^ab^	983.14 ± 62.05 ^a^
Resilience (g)	0.24 ± 0.04 ^a^	0.13 ± 0.01 ^b^	0.13 ± 0.01 ^b^	0.11 ± 0.03 ^b^	0.12 ± 0.01 ^b^

Values (means ± SDs, *n* = 5) in each row followed by different lower-case superscript letters are significantly different at *p* < 0.05 as shown by Duncan’s multiple range test.

**Table 7 foods-12-04085-t007:** Sensory characteristics and acceptability of muffins made with XanFlax powder.

Property	XanFlax Content in Mix with Refined Wheat Flour (%)				
	0	5	10	20	40
Appearance	6.08 ± 1.08 ^a^	4.89 ± 1.65 ^b^	5.19 ± 1.43 ^b^	4.74 ± 1.21 ^c^	4.54 ± 1.71 ^c^
Flavor	5.11 ± 1.21 ^a^	5.22 ± 1.27 ^a^	5.00 ± 1.59 ^ab^	4.74 ± 1.40 ^ab^	4.35 ± 1.76 ^b^
Taste	5.25 ± 1.44 ^a^	4.89 ± 1.30 ^a^	5.19 ± 1.24 ^a^	4.00 ± 1.52 ^b^	3.97 ± 1.73 ^b^
Texture	4.83 ± 1.46 ^ab^	5.08 ± 1.05 ^a^	5.17 ± 1.16 ^a^	4.56 ± 1.26 ^ab^	4.24 ± 1.81 ^b^
Overall acceptability	4.94 ± 1.60 ^ab^	5.00 ± 1.35 ^a^	4.92 ± 1.44 ^ab^	4.18 ± 1.55 ^bc^	4.06 ± 1.86 ^c^

Values (means ± SDs, *n* = 3) within rows followed by different lower-case superscript letters are significantly different at *p* < 0.05 as shown by Duncan’s multiple range test.

## Data Availability

The data from the current study are available from the corresponding author on reasonable request.

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
