# Peer review of "The Impacts of Standardized Flaxseed Meal (XanFlax) on the Physicochemical, Textural, and Sensory Properties of Muffins"

_foods, 2023, doi:10.3390/foods12224085_

Round 1

Reviewer 1 Report

Comments and Suggestions for Authors

I reviewed the manuscript titled “Physicochemical Property of a Thickener as a Novel Flaxseed Meal Formulation”. The manuscript is well written. However, the authors failed to provide more details on the XanFlax. It is not prepared by authors.

Title: title must be changed. As such, it is not reflecting the content in the manuscript.

Abstract:

Authors should revise the manuscript carefully to avoid the use of more generalized words like line 19: looked…….. evaluated could be the good word. Pleae revise entire manuscript

Methodology

How authors prepare the XanFlax. Authors say that the power was supplied by Prairie Tide Diversified Inc. Without full information, the repetition of this study is not possible.

Song et al. (2017)… citation format is not correct. Please revise it as Song et al. [9] and revise throughout the manuscript

Table 1:  what is XanFlax Powder Content (%) (0, 5, 10, 20, and 40 )?

Row 2: XanFlax powder : 0, 10, 20, 40, and 80

Wq 1: unitless can be revised

Eq 2: what is the unit of weight. This should be mentioned

2.6. Chromaticity and Section Measurements: provide citation

2.7. Texture Measurement: provide citation

Table 2: addition of power showed no weight change. Why? What happens to the weight of the powder? Powder is weightless? Zero weight?

Height is increased but no change in weight

Volume is increased but no change in weight

No change in dough yield.

Color has an influence on the product appearance. I personally prefer control groups. Convincing consumer with dark colored muffin is challenging

Results and discussion are appropriate

Conclusions should be more concise and should clear the findings and future recommendations.

References must be cross-checked 

Author Response

Thank you for your patience and recommendations for strengthening our manuscript (ID: foods- 2675623). We would like to thank the reviewers for their time and expertise in providing critical feedback in making this manuscript suitable for publication. We have revised our manuscript according to the reviewers’ comments. In addition to these changes, we have also made substantial revisions to improve the style, flow, and clarity of this manuscript. We hope these changes improve the overall quality of this manuscript for publication. We have listed the reviewers’ comments and answered them in sequence.

Reviewer #1:

Comment: I reviewed the manuscript titled “Physicochemical Property of a Thickener as a Novel Flaxseed Meal Formulation”. The manuscript is well written. However, the authors failed to provide more details on the XanFlax. It is not prepared by authors. Title: title must be changed. As such, it is not reflecting the content in the manuscript.

Response: Revised “The Impact of Standardized Flaxseed Meal (XanFlax) on Muffin Quality and Nutritional Content”

Comment: Abstract: Authors should revise the manuscript carefully to avoid the use of more generalized words like line 19: looked…….. evaluated could be the good word. Please revise entire manuscript

Response: Revised.

Comment: Methodology: How authors prepare the XanFlax. Authors say that the power was supplied by Prairie Tide Diversified Inc. Without full information, the repetition of this study is not possible.

Response: XanFlax is a trademarked product of Prairie Tide Diversified Inc., and its production is proprietary. The product is standardized, and datasheets describing XanFlax are available on request.

Comment: Song et al. (2017)… citation format is not correct. Please revise it as Song et al. [9] and revise throughout the manuscript.

Response: The year of the citation was deleted to suit the journal format.

Comment: Table 1: what is XanFlax Powder Content (%) (0, 5, 10, 20, and 40 )? Row 2: XanFlax powder: 0, 10, 20, 40, and 80

Response: Table 1 was revised. The column labels describe the % XanFlax in the flour component. As the total flour content was 200 g per formula, the added grams of XanFlax are twice the percentage.

Comment: Wq 1: unitless can be revised

Response: Revised.

Comment: Eq 2: what is the unit of weight. This should be mentioned.

Response: Revised.

Comment: 2.6. Chromaticity and Section Measurements: provide citation

Response: Bhaduri, Sikha. "A comprehensive study on physical properties of two gluten-free flour fortified muffins." Journal of Food Processing & Technology 4.7 (2013): 1-4.

Comment: 2.7. Texture Measurement: provide citation

Response:

Comment: Table 2: addition of power showed no weight change. Why? What happens to the weight of the powder? Powder is weightless? Zero weight?

Response: XanFlax was added, and an equal weight of soft wheat flour was omitted from the formula. There was no net change in weight.  

Comment: Height is increased but no change in weight

Response: The product is part gas due to the leavening effect of baking powder and cooking.

Comment: Volume is increased but no change in weight

Response: See above.

Comment: No change in dough yield.

Response: This is not that surprising, as the recipes were consistent for all muffins.

Comment: Color has an influence on the product appearance. I personally prefer control groups. Convincing consumer with dark colored muffin is challenging.

Response: Agreed. There are specific products that are dark where a dark colour is preferred. Cinnamon-rich products can be dark, as can chocolate-flavoured products.

Comment: Results and discussion are appropriate. Conclusions should be more concise and should clear the findings and future recommendations.

Response: Revised accordingly.

Comment: References must be cross-checked.

Response: Each reference was cross-checked.

Reviewer 2 Report

Comments and Suggestions for Authors

Dear Authors,

The issue of gluten-free food and its quality is very important. This is a special group of recipients for whom these foods are medicine.

The paper should include the characteristics of XanFlax powder and explain what valuable it brings to the recipe. 

At what time after baking was the moisture content determined? Why was the moisture content not determined after 24 and 48 hours of storage? This would have given a picture of the changes in the blackening of the bread. And this is a big problem in the quality of gluten-free products.

Sensory evaluation is too general and should be expanded to include additional distinctions for taste and smell, as there may be foreign flavors or foreign odors. Were there people on the panel following a gluten-free diet?

Why did hardness decrease significantly with the addition of 5% XanFlax, compared to the control sample?

For what purpose was the pH determined and what effect did it have on the quality of the muffins? What does it contribute to this work?

Conclusions should be reworded as they are statements and description of results, not conclusions.

Author Response

Physicochemical Property of a Thickener as a Novel Flaxseed Meal Formulation

Thank you for your patience and recommendations for strengthening our manuscript (ID: foods- 2675623). We would like to thank the reviewers for their time and expertise in providing critical feedback in making this manuscript suitable for publication. We have revised our manuscript according to the reviewers’ comments. In addition to these changes, we have also made substantial revisions to improve the style, flow, and clarity of this manuscript. We hope these changes improve the overall quality of this manuscript for publication. We have listed the reviewers’ comments and answered them in sequence.

Reviewer #2:

Comment: The issue of gluten-free food and its quality is very important. This is a special group of recipients for whom these foods are medicine. The paper should include the characteristics of XanFlax powder and explain what valuable it brings to the recipe. 

Response: A product sheet describing XanFlax as gluten-free is available from the supplier. Xanflax is a commercial product with a commercial label. The product has a 2-year shelf life and 5.4 g oil/30 g serving (~18% fat). The current application is not gluten-free.

Comment: At what time after baking was the moisture content determined? Why was the moisture content not determined after 24 and 48 hours of storage? This would have given a picture of the changes in the blackening of the bread. And this is a big problem in the quality of gluten-free products.

Response: As mentioned earlier, this is a gluten-free ingredient used in gluten products to replace part of the flour. We respected the reviewer's suggestion and found it interesting, but we measured the moisture content of the cooked muffins after baking and allowed them to cool at room temperature for 1 h.

Comment: Sensory evaluation is too general and should be expanded to include additional distinctions for taste and smell, as there may be foreign flavors or foreign odors. Were there people on the panel following a gluten-free diet?

Response: No one on the study panel followed a gluten-free diet. Refined soft wheat flour was used in the muffins prepared in this study. As mentioned, this is a gluten-free ingredient that is being used in a gluten product to replace a portion of the flour. In general flaxseed meal does not affect flavour [19]. 

Comment: Why did hardness decrease significantly with the addition of 5% XanFlax, compared to the control sample?

Response: When 5% of XanFlax powder was added, the volume of the muffin became smaller compared to the control, and the internal structure of the muffin became denser with the added XanFlax powder. This is likely due to the dilution of the gluten, which increased muffin hardness with the addition of XanFlax, a water-soluble gum.

Comment: For what purpose was the pH determined and what effect did it have on the quality of the muffins? What does it contribute to this work?

Response:

  1. Fat content: Flaxseed meal is higher in healthy fats, particularly omega-3 fatty acids. These fats can add a degree of tenderness to the muffins and make them less dense. This can result in a softer texture.
  2. Binding and gel formation: Flaxseed meal contains soluble fiber, which has a gelling effect when mixed with liquid. This gel formation can alter the muffin structure, preventing it from becoming too hard.
  3. Gluten reduction: Replacing a portion of the flour with a flaxseed meal reduces the gluten content in the recipe. Reducing gluten might make the muffins less likely to become hard.
  4. Nutrient interactions: The addition of flaxseed meal may influence other factors in the recipe. For instance, it can interact with other ingredients, affecting the way the muffins rise and bake. These interactions might lead to the softer texture.

Comment: Conclusions should be reworded as they are statements and description of results, not conclusions.

Response: We have summarized and updated the conclusion more briefly.

Reviewer 3 Report

Comments and Suggestions for Authors

The major corrections are recommended, since improvement in terms of chemical characterisation, evaluation of phenolic content and antioxidant activity (by different tests) of muffins is required.

Other corrections suggested for the manuscript are:

- Page 1 - Perhaps, for title it is more adequate "Physicochemical Property of a Novel Formulation with Flaxseed Meal".

- Page 1, in abstract line 16 and 33 - XanFlax could be used for preparation of "gluten-free" prodacts, but in this study tested muffins contain wheat, since the tested formulation (muffins) was not "gluten-free". Reformulate these parts of text and keywords.

- Page 1, in abstract line 24 - Please, specify is this statement for dough of for baked muffins.

- Page 1, in abstract line 25 - Add exact values.

- Page 2, line 77 - Add the sentence into the introduction section, with the enumerated nutritional compounds which are found in flaxseed meal (the by-product of the flaxseed oil extraction) and not found in the oil, with references (as stated in abstract, line 14-15).

- Page 2, line 89 - Please, specify here how this powder was produced (i.e. add some more information about this powder). Is the XanFlax powder obtained from flaxseed meal (by-product of the flaxseed oil extraction)? (In the abstract flaxseed meal is defined as a by-product, line 14 and 16)

- Page 3, line 130 - Add Equation (3) for Dough yield (%) calculation.

- Page 4, line 155 - Add trigger force and distance.

- Page 5, line 184-189 - The text above was repeated.

- Page 5, Table 2 - Check the lower-case superscript letters for Specific gravity.

- Page 5, line 190 - Check the data.

- Page 6, line 222 - Check the data.

- Page 8, line 307 - Check the value "30%".

- Page 8, line 310 and 318 - Check the reference "[37]". Is the reference "[37]" for muffins with citrus fruit?

- Page 8, line 327, 334 and 338 - Add references.

- Page 11, line 437, 445, 451 and 503 - Check the page numbers.

Author Response

Physicochemical Property of a Thickener as a Novel Flaxseed Meal Formulation

Thank you for your patience and recommendations for strengthening our manuscript (ID: foods- 2675623). We would like to thank the reviewers for their time and expertise in providing critical feedback in making this manuscript suitable for publication. We have revised our manuscript according to the reviewers’ comments. In addition to these changes, we have also made substantial revisions to improve the style, flow, and clarity of this manuscript. We hope these changes improve the overall quality of this manuscript for publication. We have listed the reviewers’ comments and answered them in sequence.

 Reviewer #3:

Comment: The major corrections are recommended, since improvement in terms of chemical characterisation, evaluation of phenolic content and antioxidant activity (by different tests) of muffins is required.

Response: Lignan is the predominant phenolic compound in flaxseed meal. This is ~2% of meal mass. The evaluation of phenolic content and the antioxidant properties of flaxseed meal are well-known and have already been published in the two studies below.

Barthet, V.J.; Klensporf-Pawlik, D.; Przybylski, R. Antioxidant activity of flaxseed meal components. Can. J. Plant Sci. 94(3):593-602

Anwar, F.; Przybylski, R. Effect of solvents extraction on total phenolics and antioxidant activity of extracts from flaxseed (Linum usitatissimum L.). Acta Sci. Pol. Technol. Aliment. 2012;11(3), 293-301

Comment: Page 1 - Perhaps, for title it is more adequate "Physicochemical Property of a Novel Formulation with Flaxseed Meal".

Response: Revised with a new title: “The Impact of Standardized Flaxseed Meal (XanFlax) on Muffin Quality and Nutritional Content”

Comment: Page 1, in abstract line 16 and 33 - XanFlax could be used for preparation of "gluten-free" prodacts, but in this study tested muffins contain wheat, since the tested formulation (muffins) was not "gluten-free". Reformulate these parts of text and keywords.

Response: We are not using XanFlax to make a gluten-free product but as a powder used to displace wheat and add functionality. Wheat contains gluten and is an important component of the dough's performance. We are discussing the dilution of gluten, not the production of a gluten-free product. We ask that the reviewer reconsider this point.

Comment: - Page 1, in abstract line 24 - Please, specify is this statement for dough of for baked muffins.

Response: The comment is regarding the muffins. The word muffins is included in 2 of 3 sentences in the identified text.

Comment: In abstract line 25 - Add exact values.

Response: The following two moisture contents were again specified: “23.71 ± 0.79% and 22.06 ± 0.30%”.

Comment: Add the sentence into the introduction section, with the enumerated nutritional compounds which are found in flaxseed meal (the by-product of the flaxseed oil extraction) and not found in the oil, with references (as stated in abstract, line 14-15).

Response: We have included the Materials and Methods section and Table 1 to describe XanFlax, a high dietary fiber product from defatted flaxseed.

Comment: Page 2, line 89 - Please, specify here how this powder was produced (i.e. add some more information about this powder). Is the XanFlax powder obtained from flaxseed meal (by-product of the flaxseed oil extraction)? (In the abstract flaxseed meal is defined as a by-product, line 14 and 16).

Response: In response to the same question as above, we inserted the Materials and Methods part and Table 1 to describe XanFlax, a high dietary fiber product from defatted flaxseed.

Comment: line 130 - Add Equation (3) for Dough yield (%) calculation.

Response: Dough yield = 100 – loss rate. Perhaps we do not need both numbers.

Comment: line 155 - Add trigger force and distance.

Response: Added analytical conditions of texture profile analyzer for trigger force and test distance.

Comment: line 184-189 - The text above was repeated.

Response: Revised

Comment: Table 2 - Check the lower-case superscript letters for Specific gravity.

Response: Revised

Comment: Page 5, line 190 - Check the data.

Response: Revised

Comment: Page 6, line 222 - Check the data

Response: Revised

Comment: Page 8, line 307 - Check the value "30%

Response: Revised

Comment: Page 8, line 310 and 318 - Check the reference "[37]". Is the reference "[37]" for muffins with citrus fruit?

Response: Revised

Comment: Page 8, line 327, 334 and 338 - Add references.

Response: According to the reviewer's comments, references were inserted, and sentences were revised and supplemented.

Comment: Page 11, line 437, 445, 451 and 503 - Check the page numbers.

Response: All four references below have been modified.

  1. Shevkani, K.; Kaur, A.; Kumar, S.; Singh, N. Cowpea protein isolates: Functional properties and application in gluten-free rice muffins. LWT Food Sci. Technol. 2015, 63, 927–933
  2. Kim, B.G.; Park, N.Y.; Lee, S.H. Quality characteristics and antioxidative activity of muffins added with coffee ground residue water extract and powder. J. Korean Soc. Food Sci. Nutr. 2016, 45(1), 76–83.
  3. Hong, J.Y. Quality characteristics of muffin added with Elaeagnus multiflora powder. Korean J. Food Preserv. 2019, 26(1), 74–82.
  4. Grigelmo-Miguel, N.; Carreras-Bolderas, E.; Martín-Belloso, O. Development of high-fruit-dietary-fiber muffins. Eur. Food Res. Technol. 1999, 210, 123–128.

Reviewer 4 Report

Comments and Suggestions for Authors

After reading the manuscript "Physicochemical Property of a Thickener as a Novel Flaxseed Meal Formulation" , I realized that the manuscript showed in some parts the scientific rigour wanted, but in other parts I have missed it.

The authors have presented critical evaluation only in some paragraphs.

The references are not exactly current, besides title,  objective must be corrected.

Thats why I have written some suggestions below in an attempt to improve the paper.

L.3- In my opinion, "muffin" should be in your title.

L.16- You mention "gluten-free" in several parts of the text, but your product is not gluten-free, since you use wheat.

So I ask you:  What is the advantage of using "XanFlax powder"? What is the innovation in your study? I don't understand why the paper emphasizes gluten-free so much. Please, clarify this in the justification for the study.

L. 31- I missed this statement in the conclusion.

L.34- Avoid words that are already in the title, it doesn't help other researchers finding your study.

L.38 - you mention "Recently" but the reference cited is from 2014.

L.59- " Flaxseed and flaxseed " - improve it, please.

L.62 - Instead of grains, maybe cereals ? Because legumes are grains as well.

L.69- Please, follow authors' guidelines of Foods Journal.

L. 74- I suggest first explaining what XanFlax is and only then reporting its applicability in other foods.

L.82- " underutilized"  Why ?  How is Xan Flax obtained ? price ?  Those who work with dysphagia know that thickeners are not cheap.

L.84- L. 83- Checking the use of " consumers preference" - it does not seem appropriate in this case. It seems to me that you've performed an acceptability test.  A lot of information is missing in the Material and methods.

L.102 - Please, provide details of all the ingredients used in the muffins formulations. MAybe in table 1. For example: Which sugar ? demerara ? refined ? butter ? which one ? salted ? unsalted ? Same for milk, salt and so on...

L.105- Which  "Cream" - why capital letters ?

L.135- Triplicate for moisture ?

L.136- Paragraph for sugar content

L.140- Paragraph for pH

L.155- "adhesiveness, springiness, cohesiveness, gumminess, chewiness, and resilence" - It wasn't reported if it was calculated, please, correct it.

L.157- Only 3 times ? Can be this a problem ? Some authors evaluate 6 or more.

L.158- The title and objective need to be evaluated about sensory analysis, please add.

For a sensory test a lot of relevant information was not included, I suggest reading papers and improving your article.  Has the project been submitted to an evaluation by a university ethics committee? Did it follow the Helsinki declaration? Please, enter the approval protocol number. Which sensory test was performed? Were they trained ? If it were, how many sessions were conducted ? What was the profile of the assessors ? How many men/women ? age of the assessors ? Are the assessors usually consumers of this product  ? Were the analyses performed in sensory booths ?   Guys, the sensory part needs to be improved, a lot of important things are missing. It can't remain like this.   L.170- Why did you choose Duncan Test? Did you try Tukey?   L.215- I am sorry, but i did not understand  these results :  0.41 ± 0.00a    0.41 ± 0.00b 0.41 ± 0.00b 0.41 ± 0.00b 0.43 ± 0.00a, check the numbers and letters.   L.223- "Previous studies have shown a direct relationship between dietary fiber content and moisture  content"  -  I don't understand why you didn't evaluate the proximate composition of the muffins. Reading this part, I think you agree. If you have this result, include it, if not, I suggest including it in the limitations of the study.   L.244- 6.88 ± 0.18c     6.71 ± 0.03bc -  I also so not understand this result, check please.   L.285- Why didn't the control treatment look as good as expected? I'm referring to the gluten net. Do you have another image? This one isn't helping.   L.299- Avoid using the 1st person plural in your text, prefer impersonal.  I suggest "It was observed", for instance.   L.302- How would you explain the negative results in this parameter? Please be clearer in the writing.   L.369-  The attributes are different in the text and in the table from those presented in the Material and methods -L.166   L.371- I think that after a table it would be interesting to include a text, perhaps in this case a discussion, limitations of the study and/or future perspectives. A table placed exactly before the conclusion is not appropriate.   L.373/ 384 /393- Avoid using the 1st person plural in your text, prefer impersonal.  I suggest "It was evaluated", for instance.   L.378- So, you need to infer in the conclusion of your paper what the best treatment was. The conclusion is too lengthy. It sounded like a summary of the study in many parts.  In others, I saw future perspectives and limitations of the study. It needs to be revised and reorganized.   Comments on the Quality of English Language

Minor editing of English language required

Author Response

Physicochemical Property of a Thickener as a Novel Flaxseed Meal Formulation

Thank you for your patience and recommendations for strengthening our manuscript (ID: foods- 2675623). We would like to thank the reviewers for their time and expertise in providing critical feedback in making this manuscript suitable for publication. We have revised our manuscript according to the reviewers’ comments. In addition to these changes, we have also made substantial revisions to improve the style, flow, and clarity of this manuscript. We hope these changes improve the overall quality of this manuscript for publication. We have listed the reviewers’ comments and answered them in sequence. 

Reviewer #4:

After reading the manuscript "Physicochemical Property of a Thickener as a Novel Flaxseed Meal Formulation, I realized that the manuscript showed in some parts the scientific rigour wanted, but in other parts I have missed it.

Comment: The references are not exactly current, besides title, objective must be corrected.

Response: The most recent references, title, and purpose have been revised.

Comment: L.3- In my opinion, "muffin" should be in your title.

Response: The title of this study was modified as follows by inserting “Muffin”: The Impact of Standardized Flaxseed Meal (XanFlax) on Muffin Quality and Nutritional Content (Lines 2, 3).

Comment: L.16- You mention "gluten-free" in several parts of the text, but your product is not gluten-free, since you use wheat. So I ask you: What is the advantage of using "XanFlax powder"? What is the innovation in your study? I don't understand why the paper emphasizes gluten-free so much. Please, clarify this in the justification for the study.

Response: The XanFlax product is gluten-free but the muffins have gluten at various levels of dilution. While we are studying the impact of the XanFlax on muffin quality we cannot ignore the impact of reducing the wheat and gluten content. It is important that the audience is aware that gluten dilution is occurring. This research marks the first time that XanFlax, a standardized flaxseed flour containing water-soluble gums, has been reported in functional baking applications.

Comment: L. 31- I missed this statement in the conclusion.

Response: Added to the conclusion of Lines 428-430.

Comment: L.34- Avoid words that are already in the title, it doesn't help other researchers finding your study.

Response: Revised (Line 34).

Comment: L.38 - you mention "Recently" but the reference cited is from 2014.

Response: Revised (Line 37).

Comment: L.59- " Flaxseed and flaxseed " - improve it, please.

Response: Revised (Line 58).

Comment: L.62 - Instead of grains, maybe cereals? Because legumes are grains as well.

Response: Revised (Line 60).

Comment: L.69- Please, follow authors' guidelines of Foods Journal.

Response: The year of the citation was deleted to suit the journal format (Line 67).

Comment: L. 74- I suggest first explaining what XanFlax is and only then reporting its applicability in other foods.

Response: Revised (Lines 72, 73).

Comment: L.82- " underutilized" Why? How is Xan Flax obtained? price? Those who work with dysphagia know that thickeners are not cheap.

Response: Revised XanFlax is press cake derived from cold pressing (note that the means by which XanFlax is standardized is proprietary information of Prairie Tide Diversified).

Comment: L.84- L. 83- Checking the use of " consumers preference" - it does not seem appropriate in this case. It seems to me that you've performed an acceptability test. A lot of information is missing in the Material and methods.

Response: Revised.

Comment: L.102 - Please, provide details of all the ingredients used in the muffins formulations. Maybe in table 1. For example: Which sugar? demerara? refined? butter? which one? salted? unsalted? Same for milk, salt and so on...

Response: Lines 99-100 and Table 1 provide details of all the ingredients used in the muffin formulations.

Comment: L.105- Which "Cream" - why capital letters?

Response: Changed to lowercase.

Comment: L.135- Triplicate for moisture?

Response: Revised

Comment: L.136- Paragraph for sugar content

Response: Revised

Comment: L.140- Paragraph for pH

Response: Revised

Comment: L.155- "adhesiveness, springiness, cohesiveness, gumminess, chewiness, and resilence" - It wasn't reported if it was calculated, please, correct it.

Response: In Table 5, the texture properties of seven muffins, including hardness, were reported.

Comment: L.157- Only 3 times? Can be this a problem? Some authors evaluate 6 or more.

Response: The tests were performed using five replicates per batch. There was sufficient statistical power to show differences in most parameters measured despite the large error terms. We agree that larger evaluation groups might have an impact. Also, the involvement of different cultural groups could impact the findings by increasing the size of the error term.

Comment: L.158- The title and objective need to be evaluated about sensory analysis, please add.

Response: Revised and Added.

Comment: For a sensory test a lot of relevant information was not included, I suggest reading papers and improving your article. Has the project been submitted to an evaluation by a university ethics committee? Did it follow the Helsinki declaration? Please, enter the approval protocol number. Which sensory test was performed? Were they trained? If it were, how many sessions were conducted? What was the profile of the assessors? How many men/women? age of the assessors? Are the assessors usually consumers of this product? Were the analyses performed in sensory booths? Guys, the sensory part needs to be improved, a lot of important things are missing. It can't remain like this.

Response: We have already inserted in the “Institutional Review Board Statement” section on lines 568-570 the submission documents and approval protocol number of the sensory analysis that this study was submitted to for evaluation by the university ethics committee. In addition, we already mentioned that we conducted five attributes (appearance, flavor, taste, texture, and overall acceptability) on 35 food and nutrition majors from Gangseo University who were trained in sensory analysis. Reviewer 3's additional comments have been inserted in Section 2.8 below.

“Sensory profiling was conducted by thirty-five skilled Korean panelists (25 women and 10 men, mean age 22 years) majoring in food and nutrition at Gangseo University, who were familiar with the characteristics and sensory evaluation method of muffins containing XanFlax. The panelists selected had the ability to perform sensory tasks and had interacted in discussions of sensory attributes during a 10-min briefing session conducted to train the assessors to be familiar with the muffins‘ products and the prescribed sensory evaluation procedures. The panelists were asked to rate their acceptability using a 7-point scale, with 1 indicating "dislike very much", 4 indicating "average", and 7 indicating "like very much". The muffins were cut into 2 × 2 × 2 cm pieces and served on a white plate, and participants were instructed to rinse their mouths with water after trying each sample. The acceptance test evaluated five attributes: appearance, flavor, taste, texture, and overall acceptability [25].”

Comment: L.170- Why did you choose Duncan Test? Did you try Tukey?

Response: When the number of samples is the same, statistical similarity was derived using the Duncan method, which compares adjacent average values step by step, rather than Tukey, which is the most used test technique.

Comment: L.215- I am sorry, but i did not understand these results: 0.41 ± 0.00a 0.41 ± 0.00b 0.41 ± 0.00b 0.41 ± 0.00b 0.43 ± 0.00a, check the numbers and letters.

Response: The superscript of the value of the control group (without XanFlax powder added) was modified.

Comment: L.223- "Previous studies have shown a direct relationship between dietary fiber content and moisture content" - I don't understand why you didn't evaluate the proximate composition of the muffins. Reading this part, I think you agree. If you have this result, include it, if not, I suggest including it in the limitations of the study.

Response: Dietary fibre was not determined by proximate analysis. The soluble dietary fibre of XanFlax is 30%, while refined soft wheat flour (white) is ~1%

Comment: L.244- 6.88 ± 0.18c  6.71 ± 0.03bc - I also so not understand this result, check please.

Response: The superscripts of the value of the two groups were modified.

Comment: L.285- Why didn't the control treatment look as good as expected? I'm referring to the gluten net. Do you have another image? This one isn't helping.

Response: We have other images.

Comment: L.299- Avoid using the 1st person plural in your text, prefer impersonal. I suggest "It was observed", for instance.

Response: Revised.

Comment: L.302- How would you explain the negative results in this parameter? Please be clearer in the writing.

Response: Addressed.

  1. Reduced Gluten Content: Refined wheat flour contains gluten, which is a protein responsible for the elasticity and adhesiveness in baked goods. Flaxseed meal, on the other hand, does not contain gluten. By replacing some of the wheat flour with flaxseed meal, you are reducing the overall gluten content in the muffin batter. This can lead to a less sticky and adhesive texture in the final product.
  2. Increased Fat Content: Flaxseed meal is relatively high in healthy fats, particularly omega-3 fatty acids. These fats can have a lubricating effect in the batter, reducing the stickiness and making it less adhesive compared to a batter made solely with wheat flour. The fats can also contribute to a softer texture in the finished product.
  3. Absorption of Moisture: Flaxseed meal has the ability to absorb a significant amount of liquid. When incorporated into the recipe, it may absorb some of the moisture that would otherwise contribute to the adhesiveness of the muffins. This absorption can result in a drier texture and reduce stickiness.

Comment: L.369- The attributes are different in the text and in the table from those presented in the Material and methods

Response: We made the same modifications compared to the Materials and Methods section.

Comment: L.166 L.371- I think that after a table, it would be interesting to include a text, perhaps in this case a discussion, limitations of the study and/or future perspectives. A table placed exactly before the conclusion is not appropriate.

Response: Addressed by expecting conclusions and future prospects for research

Comment: L.373/ 384 /393- Avoid using the 1st person plural in your text, prefer impersonal. I suggest "It was evaluated", for instance.

Response: Revised.

Comment: L.378- So, you need to infer in the conclusion of your paper what the best treatment was. The conclusion is too lengthy. It sounded like a summary of the study in many parts. In others, I saw future perspectives and limitations of the study. It needs to be revised and reorganized.

Response: Following Reviewer 3's comment, we have summarized and updated the conclusion more briefly.

Comment: Minor editing of English language required.

Response: The English editing was revised again as requested by Reviewer 3.

Round 2

Reviewer 3 Report

Comments and Suggestions for Authors

The corrections are still required for the manuscript. For title, it is more adequate "The Impact of Flaxseed Meal on Muffin Physicochemical, Textural and Sensory Properties". The term "Nutritional Content" is not adequate since fats, proteins,..are not determined for prepared muffins.

Other corrections for the manuscript are:

- Page 1, line 32 - Change "with the 10% XanFlax treatment" to "with the 5% and 10% XanFlax treatments".

- Page 3, line 121 - Add ".".

- Page 4, line 128 - Add ".".

- Page 4, line 134 - Add Equation (3) for Dough yield (%) calculation or change "Equations (1–3)" to "Equations (1) and (2)" and in the Table 3. delete  Dough yield row.

- Page 4, line 156 - Add reference Ullah et al. (2019) in the references list.

- Page 5, line 195-200 - Again, the text above was repeated.

- Page 5, line 202 - Add comment for the volume of the muffins with 5% and 10% of XanFlax powder.

- Page 6, line 233 - Add "22.35 ± 0.60%  in the 20% group", as all other values are given in the text.

- Page 12 - References No. 12 and No. 15 are the same. For this, the corrections are required in the text and in the references list.

Author Response

Thank you for your patience and recommendations for strengthening our manuscript (ID: foods- 2675623). We would like to thank the reviewers for their time and expertise in providing critical feedback in making this manuscript suitable for publication. We have revised our manuscript according to the reviewers’ comments. In addition to these changes, we have also made substantial revisions to improve the style, flow, and clarity of this manuscript. We hope these changes improve the overall quality of this manuscript for publication. We have listed the reviewers’ comments and answered them in sequence.

Reviewer #3:

Comment: The corrections are still required for the manuscript. For title, it is more adequate "The Impact of Flaxseed Meal on Muffin Physicochemical, Textural and Sensory Properties". The term "Nutritional Content" is not adequate since fats, proteins,..are not determined for prepared muffins.

Response: Revised (Line 3).

Comment: Page 1, line 32 - Change "with the 10% XanFlax treatment" to "with the 5% and 10% XanFlax treatments".

Response: Corrected (Line 32).

Comment: Page 3, line 121 - Add ".".

Response: A dot (.) was inserted (Line 121).

Comment: Page 4, line 128 - Add ".".

Response: A dot (.) was inserted (Line 128).

Comment: Page 4, line 134 - Add Equation (3) for Dough yield (%) calculation or change "Equations (1–3)" to "Equations (1) and (2)" and in the Table 3. delete Dough yield row.

Response: Removed all “Dough yield” (Lines 134, 227).

Comment: Page 4, line 156 - Add reference Ullah et al. (2019) in the references list (Lines 156, 508, 509).

Response: Replaced it with another reference and inserted it into the reference list.

Comment: Page 5, line 195-200 - Again, the text above was repeated.

Response: Duplicate sentences were removed (Lines 200-203).

Comment: Page 5, line 202 - Add comment for the volume of the muffins with 5% and 10% of XanFlax powder.

Response: Inserted (Lines 200-203).

Comment: Page 6, line 233 - Add "22.35 ± 0.60% in the 20% group", as all other values are given in the text.

Response: Inserted (Line 234).

Comment: Page 12 - References No. 12 and No. 15 are the same. For this, the corrections are required in the text and in the references list.

Response: Numbers 12 and 15 in the reference list (Lines 482, 483, 488, 489) have been modified as follows.

[12] Shim, Y.Y.; Kim, J.H.; Cho, J.Y.; Reaney, M.J.T. Health benefits of flaxseed and its peptides (linusorbs). Crit. Rev. Food Sci. Nutr. 2022, 1–20. https://doi.org/10.1080/10408398.2022.2119363.

[15] Kajla, P.; Sharma, A.; Sood, D. R. Flaxseed-a potential functional food source. J. Food Sci. Technol. 2015, 52(4), 1857–1871. https://doi: 10.1007/s13197-014-1293-y.

Reviewer 4 Report

Comments and Suggestions for Authors After another evaluation of the manuscript, I realized  improvement in the quality of the paper. The authors have acceptedsome of my request. It is always useful to ask a native speaker for a final appreciation. They improved methodology and corrected tables.   I still have some more considerations :

Flaxseed, XanFlax; muffin - remain in the title and keywords

I still don't see the point in emphasizing gluten-free in your paper, remembering that gluten is only a problem for people who have gluten-related diseases and that wheat is still our gold standard technologically and sensorially for pasta, especially bread.  I've done research with celiacs and patients with RGDs and these patients are well-informed, including scientific reading, and they know that  flaxseed-based products are gluten-free.

L.118- Salt continued without detailing which salt was used

L.119-  (0 5 10 20 40 is in the tables and footnotes as well ) Do they need to be twice ?  I mean in the tables and in the footnotes?- check all the tables as well

L.169 - "The panelists selected had the ability to perform sensory"

Were they trained? 

Which sensory test was performed?

If it were, how many sessions were conducted?

What was the profile of the assessors? How many men/women? age of the assessors? Are the assessors usually consumers of this product? Were the analyses performed in sensory booths? 

L.156- Insert reference to Foods rules

Comments on the Quality of English Language

Minor editing of English language required.

Author Response

Reviewer #4:

After another evaluation of the manuscript, I realized improvement in the quality of the paper. The authors have accepted some of my request. It is always useful to ask a native speaker for a final appreciation. They improved methodology and corrected tables. I still have some more considerations:

Comment: Flaxseed, XanFlax; muffin - remain in the title and keywords.

Response: Remained.

Comment: I still don't see the point in emphasizing gluten-free in your paper, remembering that gluten is only a problem for people who have gluten-related diseases, and that wheat is still our gold standard technologically and sensorially for pasta, especially bread. I've done research with celiacs and patients with RGDs and these patients are well-informed, including scientific reading, and they know that flaxseed-based products are gluten-free.

Response: While it is true that gluten is not a concern for individuals without gluten-related diseases, we would like to clarify that our rationale is based on flour functionality. Gluten is a highly functional ingredient that produces desirable characteristics of baked leavened products. If we displace wheat flour and its functional ingredient, gluten, with flaxseed flour, we are looking at the functional aspects, and this has no relevance to individuals with celiac disease. We understand the Reviewer’s experience with celiac patients and those with gluten-related diseases, and we respect their knowledge and preferences. Those individuals would be able to recognize that the products we are investigating are not intended for their consumption. However, our research addresses a broader audience, including those who may be interested in the impact of flaxseed-based products on muffins with lower flour content and increased flaxseed meal content.

Comment: L.118- Salt continued without detailing which salt was used.

Response: Refined salt was used (Line 119).

Comment: L.119- (0 5 10 20 40 is in the tables and footnotes as well) Do they need to be twice ? I mean in the tables and in the footnotes?- check all the tables as well.

Response: All other groups were deleted except the control group (Line 120).

Comment: L.169 – “The panelists selected had the ability to perform sensory".

Were they trained? Which sensory test was performed? If it were, how many sessions were conducted?

Response: The text has been modified to reflect the reviewer's comment as shown below.

Lines 171-175: “The assessors selected majoring in food and nutrition had the ability to perform sensory tasks and had interacted in discussions of sensory attributes during six 1 h training sessions conducted to train the assessors to be familiar with the muffins‘ products and the prescribed sensory evaluation procedures related to taste and tactile sensations. “

Comment: What was the profile of the assessors? How many men/women? age of the assessors? Are the assessors usually consumers of this product? Were the analyses performed in sensory booths?

Response: The authors stated that the assessors were students majoring in food and nutrition, and their gender, age, and total number of evaluators were already specified as follows.

Lines 166-177: “A total of 35 assessors (25 women and 10 men) aged between 21 and 24 years (mean age: 22 ± 2 years) were selected from students of the Department of Food and Nutrition of Gangseo University. Only subjects who were regular muffin consumers (muffin consumption corresponding to 1–2 muffins per week) were involved in the study. None of the participants had previous or present taste or smell disorders. Informed consent was obtained from all subjects. The assessors selected majoring in food and nutrition had the ability to perform sensory tasks and had interacted in discussions of sensory attributes during six 1 h training sessions conducted to train the assessors to be familiar with the muffins‘ products and the prescribed sensory evaluation procedures related to taste and tactile sensations. The training sessions were performed in a collective room and sensory booths of the Department of Food and Nutrition (Gangseo University), designed under ISO guidelines [25].”

Comment: L.156- Insert reference to Foods rules

Response: Reference 24 has been inserted (Lines 508, 509).

[24] Yoon, J.A. Quality characteristics of muffins added with Suaeda japonica powder. J. East Asian Soc. Dietary Life 2021, 31(5), 311–319